# LEARNING EFFECTIVE EXPLORATION STRATEGIES FOR CONTEXTUAL BANDITS

## ABSTRACT

In contextual bandits, an algorithm must choose actions given observed contexts, learning from a reward signal that is observed only for the action chosen. This leads to an exploration/exploitation trade-off: the algorithm must balance taking actions it already believes are good with taking new actions to potentially discover better choices. We develop a meta-learning algorithm, MÊLÉE, that learns an exploration policy based on simulated, synthetic contextual bandit tasks. MÊLÉE uses imitation learning against these simulations to train an exploration policy that can be applied to true contextual bandit tasks at test time. We evaluate on both a natural contextual bandit problem derived from a learning to rank dataset as well as hundreds of simulated contextual bandit problems derived from classification tasks. MÊLÉE outperforms seven strong baselines on most of these datasets by leveraging a rich feature representation for learning an exploration strategy.

## 1  INTRODUCTION

In a contextual bandit problem, an agent attempts to optimize its behavior over a sequence of rounds based on limited feedback (Kaelbling, 1994; Auer, 2003; Langford & Zhang, 2008). In each round, the agent chooses an action based on a context (features) for that round, and observes a reward for that action but no others (§2). Contextual bandit problems arise in many real-world settings like online recommendations and personalized medicine. As in reinforcement learning, the agent must learn to balance *exploitation* (taking actions that, based on past experience, it believes will lead to high instantaneous reward) and *exploration* (trying actions that it knows less about).

In this paper, we present a meta-learning approach to automatically learn a good exploration mechanism from data. To achieve this, we use synthetic supervised learning data sets on which we can simulate contextual bandit tasks in an offline setting. Based on these simulations, our algorithm, MÊLÉE (MEta LEarner for Exploration)[1], learns a good heuristic exploration strategy that should ideally generalize to future contextual bandit problems. MÊLÉE contrasts with more classical approaches to exploration (like $\epsilon$-greedy or LinUCB; see §6), in which exploration strategies are constructed by expert algorithm designers. These approaches often achieve provably good exploration strategies in the worst case, but are potentially overly pessimistic and are sometimes computationally intractable.

At training time (§3.2), MÊLÉE simulates many contextual bandit problems from fully labeled synthetic data. Using this data, in each round, MÊLÉE is able to counterfactually simulate what would happen under all possible action choices. We can then use this information to compute regret estimates for each action, which can be optimized using the AggreVaTe imitation learning algorithm (Ross & Bagnell, 2014). Our imitation learning strategy mirrors that of the meta-learning approach of Bachman et al. (2017) in the active learning setting. We present a simplified, stylized analysis of the behavior of MÊLÉE to ensure that our cost function encourages good behavior (§4). Empirically, we use MÊLÉE to train an exploration policy on only synthetic datasets and evaluate this policy on both a contextual bandit task based on a natural learning to rank dataset as well as three hundred simulated contextual bandit tasks (§5.2). We compare the trained policy to a number of alternative exploration algorithms, and show the efficacy of our approach (§5.3).

---

[1]**Code release:** the code is available online `https://www.dropbox.com/sh/dc3v8po5cbu8zaw/AACu1f_4c4wIZxD1e7W0KVZ0a?dl=0`

## 2 PRELIMINARIES: CONTEXTUAL BANDITS AND POLICY OPTIMIZATION

Contextual bandits is a model of interaction in which an agent chooses actions (based on contexts) and receives immediate rewards for that action alone. For example, in a simplified news personalization setting, at each time step $t$, a user arrives and the system must choose a news article to display to them. Each possible news article corresponds to an action $a$, and the user corresponds to a context $x_t$. After the system chooses an article $a_t$ to display, it can observe, for instance, the amount of time that the user spends reading that article, which it can use as a reward $r_t(a_t)$.

Formally, we largely follow the setup and notation of Agarwal et al. (2014). Let $\mathcal{X}$ be an input space of contexts (users) and $[K] = \{1, \ldots, K\}$ be a finite action space (articles). We consider the statistical setting in which there exists a fixed but unknown distribution $\mathcal{D}$ over pairs $(x, \boldsymbol{r}) \in \mathcal{X} \times [0,1]^K$, where $\boldsymbol{r}$ is a vector of rewards (for convenience, we assume all rewards are bounded in $[0,1]$). In this setting, the world operates iteratively over rounds $t = 1, 2, \ldots$. Each round $t$:

1. The world draws $(x_t, \boldsymbol{r}_t) \sim \mathcal{D}$ and reveals context $x_t$.
2. The agent (randomly) chooses action $a_t \in [K]$ based on $x_t$, and observes reward $r_t(a_t)$.

The goal of an algorithm is to maximize the cumulative sum of rewards over time. Typically the primary quantity considered is the *average regret* of a sequence of actions $a_1, \ldots, a_T$ to the behavior of the best possible function in a prespecified class $\mathcal{F}$:

$$\lambda(a_1, \ldots, a_T) = \max_{f \in \mathcal{F}} \frac{1}{T} \sum_{t=1}^{T} \left[ r_t(f(x_t)) - r_t(a_t) \right] \tag{1}$$

An agent is call *no-regret* if its average regret is zero in the limit of large $T$.

To produce a good agent for interacting with the world, we assume access to a function class $\mathcal{F}$ and to an *oracle policy optimizer* for that function class. For example, $\mathcal{F}$ may be a set of single layer neural networks mapping user features $x \in \mathcal{X}$ to predicted rewards for actions $a \in [K]$. Formally, the observable record of interaction resulting from round $t$ is the tuple $(x_t, a_t, r_t(a_t), p_t(a_t)) \in \mathcal{X} \times [K] \times [0,1] \times [0,1]$, where $p_t(a_t)$ is the probability that the agent chose action $a_t$, and the full history of interaction is $h_t = \langle (x_i, a_i, r_i(a_i), p_i(a_i)) \rangle_{i=1}^{t}$. The oracle policy optimizer, POLOPT, takes as input a *history* of user interactions and outputs an $f \in \mathcal{F}$ with low expected regret.

A standard example of a policy optimizer is to combine inverse propensity scaling (IPS) with a regression algorithm (Dudik et al., 2011). Here, given a history $h$, each tuple $(x, a, r, p)$ in that history is mapped to a multiple-output regression example. The input for this regression example is the same $x$; the output is a vector of $K$ costs, all of which are zero except the $a^{\text{th}}$ component, which takes value $r/p$. This mapping is done for all tuples in the history, and a supervised learning algorithm on the function class $\mathcal{F}$ is used to produce a low-regret regressor $f$. This is the function returned by the policy optimizer. IPS, and other estimators that have lower-variance than IPS (such as the doubly-robust estimator), have the property of being unbiased. In experiments, we use the direct method (Dudik et al., 2011) largely for its simplicity, however, MÊLÉE is agnostic to the type of the estimator used by the policy optimizer.

## 3 APPROACH: LEARNING AND EFFECTIVE EXPLORATION STRATEGY

In order to have an effective approach to the contextual bandit problem, one must be able to both optimize a policy based on historic data and make decisions about how to explore. The exploration/exploitation dilemma is fundamentally about long-term payoffs: is it worth trying something potentially suboptimal *now* in order to learn how to behave better in the future? A particularly simple and effective form of exploration is $\epsilon$-greedy: given a function $f$ output by POLOPT, act according to $f(x)$ with probability $(1 - \epsilon)$ and act uniformly at random with probability $\epsilon$. Intuitively, one would hope to improve on such a strategy by taking more (any!) information into account; for instance, basing the probability of exploration on $f$'s uncertainty. In this section, we describe MÊLÉE, first by describing how it operates at test time when applied to a new contextual bandit problem (§3.1), and then by describing how to train it using synthetic simulated contextual bandit problems (§3.2).

## 3.1 Test Time Behavior of Mêlée

Our goal in this paper is to *learn* how to explore from experience. The training procedure for MÊLÉE will use offline supervised learning problems to learn an *exploration policy* $\pi$, which takes *two inputs*: a function $f \in \mathcal{F}$ and a context $x$, and outputs an action. In our example, $f$ will be the output of the policy optimizer on all historic data, and $x$ will be the current user. This is used to produce an agent which interacts with the world, maintaining an initially empty history buffer $h$, as:

1. The world draws $(x_t, \boldsymbol{r}_t) \sim \mathcal{D}$ and reveals context $x_t$.
2. The agent computes $f_t \leftarrow \text{POLOPT}(h)$ and a greedy action $\tilde{a}_t = \pi(f_t, x_t)$.
3. The agent plays $a_t = \tilde{a}_t$ with probability $(1 - \mu)$, and $a_t$ uniformly at random otherwise.
4. The agent observes $r_t(a_t)$ and appends $(x_t, a_t, r_t(a_t), p_t)$ to the history $h$, where $p_t = \mu/K$ if $a_t \neq \tilde{a}_t$; and $p_t = 1 - \mu + \mu/K$ if $a_t = \tilde{a}_t$.

Here, $f_t$ is the function optimized on the historical data, and $\pi$ uses it and $x_t$ to choose an action. Intuitively, $\pi$ might choose to use the prediction $f_t(x_t)$ most of the time, unless $f_t$ is quite uncertain on this example, in which case $\pi$ might choose to return the second (or third) most likely action according to $f_t$. The agent then performs a small amount of additional $\mu$-greedy-style exploration: most of the time it acts according to $\pi$ but occasionally it explores some more. In practice (§5), we find that setting $\mu = 0$ is optimal in aggregate, but non-zero $\mu$ is necessary for our theory (§4).

Importantly, we wish to train $\pi$ using one set of tasks (for which we have fully supervised data on which to run simulations) and apply it to wholly different tasks (for which we only have bandit feedback). To achieve this, we allow $\pi$ to depend representationally on $f_t$ in arbitrary ways: for instance, it might use features that capture $f_t$'s uncertainty on the current example (see §5.1 for details). We additionally allow $\pi$ to depend in a *task-independent* manner on the history (for instance, which actions have not yet been tried): it can use features of the actions, rewards and probabilities in the history but *not* depend directly on the contexts $x$. This is to ensure that $\pi$ only learns to explore and not also to solve the underlying task-dependent classification problem. Because $\pi$ needs to learn to be task independent, we found that if $f_t$'s predictions were uncalibrated, it was very difficult for $\pi$ to generalize well to unseen tasks. Therefore, we additionally allow $\pi$ to depend on a *very small* amount of fully labeled data from the task at hand, which we use to allow $\pi$ to calibrate $f_t$'s predictions. In our experiments we use only 30 fully labeled examples, but alternative approaches to calibrating $f_t$ that do not require this data would be preferable.

## 3.2 Training Mêlée by Imitation Learning

The meta-learning challenge is: how do we learn a good exploration policy $\pi$? We assume we have access to *fully labeled* data on which we can train $\pi$; this data must include context/reward pairs, but where the reward for *all* actions is known. This is a weak assumption: in practice, we use purely synthetic data as this training data; one could alternatively use any fully labeled classification dataset (Beygelzimer & Langford, 2009). Under this assumption about the data, it is natural to think of $\pi$'s behavior as a sequential decision making problem in a simulated setting, for which a natural class of learning algorithms to consider are imitation learning algorithms (Daumé et al., 2009; Ross et al., 2011; Ross & Bagnell, 2014; Chang et al., 2015).[2]

Informally, at training time, MÊLÉE will treat one of these synthetic datasets as if it were a contextual bandit dataset. At each time step $t$, it will compute $f_t$ by running POLOPT on the historical data, and then ask: for *each* action, what would the long time reward look like if I were to take this action. Because the training data for MÊLÉE is fully labeled, this can be evaluated for each possible action, and a policy $\pi$ can be learned to maximize these rewards. More formally, in imitation learning, we assume training-time access to an *expert*, $\pi^\star$, whose behavior we wish to learn to imitate at test-time. From this, we can define an optimal reference policy $\pi^\star$, which effectively "cheats" at training time by looking at the true labels. The learning problem is then to estimate $\pi$ to have as similar behavior to $\pi^\star$ as possible, but without access to those labels.

The imitation learning algorithm we use is AggreVaTe (Ross & Bagnell, 2014) (closely related to DAgger (Ross et al., 2011)), and is instantiated for the contextual bandits meta-learning problem in Alg 1. AggreVaTe learns to choose actions to minimize the cost-to-go of the expert rather than the

---

[2]In other work on meta-learning, such problems are often cast as full *reinforcement-learning* problems. We opt for imitation learning instead because it is computationally attractive and effective when a simulator exists.

---

**Algorithm 1** MÊLÉE (supervised training sets $\{S_m\}$, hypothesis class $\mathcal{F}$, exploration rate $\mu$, number of validation examples $N_{Val}$, feature extractor $\Phi$)

1: **for** round $n = 1, 2, \ldots, N$ **do**
2:     initialize meta-dataset $D = \{\}$, choose $S$ at random from $\{S_m\}$, and set history $h_0 = \{\}$
3:     partition and permute $S$ randomly into train $Tr$ and validation $Val$ where $|Val| = N_{Val}$
4:     **for** round $t = 1, 2, \ldots, |Tr|$ **do**
5:         let $(x_t, \boldsymbol{r}_t) = Tr_t$
6:         **for** each action $a = 1, \ldots, K$ **do**
7:             optimize $f_{t,a} = \text{POLOPT}(\mathcal{F}, h_{t-1} \oplus (x_t, a, r_t(a), 1 - \frac{K-1}{K}\mu))$ on augmented history
8:             roll-out: estimate $\hat{c}_a$, the cost-to-go of $a$, using $r_t(a)$ and a roll-out policy $\pi^{\text{out}}$ on $f_{t,a}$
9:         **end for**
10:        compute $f_t = \text{POLOPT}(\mathcal{F}, h_{t-1})$
11:        aggregate $D \leftarrow D \oplus (\Phi(f_t, x_t, h_{t-1}, Val), \langle \hat{c}_1, \ldots, \hat{c}_K \rangle)$
12:        roll-in: $a_t \sim \frac{\mu}{K}\mathbf{1}_K + (1 - \mu)\pi_{n-1}(f_t, x_t)$ with probability $p_t$, where $\mathbf{1}$ is the ones-vector
13:        append history $h_t \leftarrow h_{t-1} \oplus (x_t, a_t, r_t(a_t), p_t)$
14:     **end for**
15:     update $\pi_n = \text{LEARN}(D)$
16: **end for**
17: **return** $\{\pi_n\}_{n=1}^N$

---

zero-one classification loss of mimicking its actions. On the first iteration AggreVaTe collects data by observing the expert perform the task, and in each trajectory, at time $t$, explores an action $a$ in state $s$, and observes the cost-to-go $Q$ of the expert after performing this action.

Following the AggreVaTe template, MÊLÉE operates in an iterative fashion, starting with an arbitrary $\pi$ and improving it through interaction with an expert. Over $N$ rounds, MÊLÉE selects random training sets and simulates the test-time behavior on that training set. The core functionality is to generate a number of states $(f_t, x_t)$ on which to train $\pi$, and to use the supervised data to estimate the value of every action from those states. MÊLÉE achieves this by sampling a random supervised training set and setting aside some validation data from it (line 3). It then simulates a contextual bandit problem on this training data; at each time step $t$, it tries *all* actions and "pretends" like they were appended to the current history (line 7) on which it trains a new policy and evaluates it's **roll-out value** (line 8). This yields, for each $t$, a new training example for $\pi$, which is added to $\pi$'s training set (line 11); the features for this example are features of the classifier based on true history (line 10) (and possibly statistics of the history itself), with a label that gives, for each action, the corresponding cost-to-go of that action (the $c_a$s computed in line 8). MÊLÉE then must commit to a **roll-in action** to *actually* take; it chooses this according to a roll-in policy (line 12). MÊLÉE has no explicit "exploitation policy", exploitation happens when $\pi$ chooses the same action as $f_t$, while exploration happens when it chooses a different action. In learning to explore, MÊLÉE simultaneously learns when to exploit.

**Roll-in actions.** The distribution over states visited by MÊLÉE depends on the actions taken, and in general it is good to have that distribution match what is seen at test time. This distribution is determined by a *roll-in* policy (line 12), controlled in MÊLÉE by exploration parameter $\mu \in [0, 1]$. As $\mu \to 1$, the roll-in policy approaches a uniform random policy; as $\mu \to 0$, the roll-in policy becomes deterministic. When the roll-in policy does not explore, it acts according to $\pi(f_t, .)$.

**Roll-out values.** The ideal value to assign to an action (from the perspective of the imitation learning procedure) is that total reward (or advantage) that would be achieved in the long run if we took this action and then behaved according to our final learned policy. Unfortunately, during training, we do not yet know the final learned policy. Thus, a surrogate roll-out policy $\pi^{\text{out}}$ is used instead. A convenient, and often computationally efficient alternative, is to evaluate the value assuming all future actions were taken by the expert (Langford & Zadrozny, 2005; Daumé et al., 2009; Ross & Bagnell, 2014). In our setting, at any time step $t$, the expert has access to the fully supervised reward vector $\boldsymbol{r}_t$ for the context $\boldsymbol{x}_t$. When estimating the roll-out value for an action $a$, the expert will return the true reward value for this action $r_t(a)$ and we use this as our estimate for the roll-out value.

## 4 THEORETICAL GUARANTEES

We analyze MÊLÉE, showing that the no-regret property of AGGREVATE can be leveraged in our meta-learning setting for learning contextual bandit exploration. In particular, we first relate the regret of the learner in line 15 to the overall regret of $\pi$. This will show that, *if the underlying classifier improves sufficiently quickly, MÊLÉE will achieve sublinear regret*. We then show that for a specific choice of underlying classifier (BANDITRON), this is achieved. MÊLÉE is an instantiation of AGGREVATE (Ross & Bagnell, 2014); as such, it inherits AGGREVATE's regret guarantees.

**Theorem 1 (Thm 2.2 of Ross & Bagnell (2014), adapted)** *After $N$ rounds, if LEARN (line 15) is no-regret algorithm, then as $N \to \infty$, with probability 1, it holds that $J(\bar{\pi}) \leq J(\pi^\star) + 2T\sqrt{K\hat{\epsilon}_{class}(T)}$, where $J(\cdot)$ is the reward of the exploration policy, $\bar{\pi}$ is the average policy returned, and $\hat{\epsilon}_{class}(T)$ is the average regression regret for each $\pi_n$ accurately predicting $\hat{c}$, where*

$$\hat{\epsilon}_{class}(T) = \min_{\pi \in \Pi} \frac{1}{N} \hat{\mathbb{E}}_{t \sim U(T), s \sim d_{\pi_i}^t} \sum_{i=1}^{N} \left[ Q_{T-t+1}^\star(s, \pi) - \min_a Q_{T-t+1}^\star(s, a) \right] \quad (2)$$

*is the empirical minimum expected cost-sensitive classification regret achieved by policies in the class $\Pi$ on all the data over the $N$ iterations of training when compared to the Bayes optimal regressor, for $U(T)$ the uniform distribution over $\{1, \ldots, T\}$, $d_\pi^t$ the distribution of states at time $t$ induced by executing policy $\pi$, and $Q^\star$ the cost-to-go of the imitation learning expert.*

Thus, achieving low regret at the problem of learning $\pi$ on the training data it observes ("$D$" in MÊLÉE), i.e. $\hat{\epsilon}_{class}(T)$ is small, translates into low regret in the contextual-bandit setting. At first glance this bound looks like it may scale linearly with $T$. However, the bound in Theorem 1 is dependent on $\hat{\epsilon}_{class}(T)$. Note however, that $s$ is a combination of the context vector $x_t$ and the classification function $f_t$. As $T \to \infty$, one would hope that $f_t$ improves significantly and $\hat{\epsilon}_{class}(T)$ decays quickly. Thus, sublinear regret may still be achievable when $f$ learns sufficiently quickly as a function of $T$. For instance, if $f$ is optimizing a strongly convex loss function, online gradient descent achieves a regret guarantee of $O(\frac{\log T}{T})$ (Hazan et al., 2016, Theorem 3.3), potentially leading to a regret for MÊLÉE of $O(\sqrt{(\log T)/T})$.

The above statement is informal (it does not take into account the interaction between learning $f$ and $\pi$). However, we can show a specific concrete example: we analyze MÊLÉE's test-time behavior when the underlying learning algorithm is BANDITRON. BANDITRON is a variant of the multiclass Perceptron that operates under bandit feedback. Details of this analysis (and proofs, which directly follow the original BANDITRON analysis) are given in Appendix A; here we state the main result. Let $\gamma_t = \Pr[r_t(\pi(f_t, x_t) = 1)|x_t] - \Pr[r_t(f_t(x_t)) = 1|x_t]$ be the edge of $\pi(f_t, .)$ over $f$, and let $\Gamma = \frac{1}{T} \sum_{t=1}^{T} \mathbb{E}\frac{1}{1+K\gamma_t}$ be an overall measure of the edge. For instance if $\pi$ simply returns $f$'s prediction, then all $\gamma_t = 0$ and $\Gamma = 1$. We can then show the following:

**Theorem 2** *Assume that for the sequence of examples, $(x_1, \boldsymbol{r}_1), (x_2, \boldsymbol{r}_2), \ldots, (x_T, \boldsymbol{r}_T)$, we have, for all $t$, $||x_t|| \leq 1$. Let $W^\star$ be any matrix, let $L$ be the cumulative hinge-loss of $W^\star$, let $\mu$ be a uniform exploration probability, and let $D = 2||W^\star||_F^2$ be the complexity of $W^\star$. Assume that $\mathbb{E}\gamma_t \geq 0$ for all $t$. Then the number of mistakes $M$ made by MÊLÉE with BANDITRON as POLOPT satisfies:*

$$\mathbb{E}M \leq L + K\mu T + 3\max\left\{ D\Gamma/\mu, \sqrt{DTK\Gamma\mu} \right\} + \sqrt{DL\Gamma/\mu} \quad (3)$$

*where the expectation is taken with respect to the randomness of the algorithm.*

Note that under the assumption $\mathbb{E}\gamma_t \geq 0$ for all $t$, we have $\Gamma \leq 1$. The analysis gives the same mistake bound for BANDITRON but with the factor of $\Gamma$, hence this result improves upon the BANDITRON analysis only when $\Gamma < 1$ (in the realizable setting, the number of mistakes is analogous to the regret). This result is highly stylized and the assumption that $\mathbb{E}\gamma_t \geq 0$ is overly strong. This assumption ensures that $\pi$ never decreases the probability of a "correct" action. It does, however, help us understand the behavior of MÊLÉE, qualitatively: First, the quantity that matters in Theorem 2, $\mathbb{E}_t\gamma_t$ is (in the 0/1 loss case) exactly what MÊLÉE is optimizing: the expected improvement for choosing an action against $f_t$'s recommendation. Second, the benefit of using $\pi$ within BANDITRON is a *local* benefit: because $\pi$ is trained with expert rollouts, as discussed in §4, the primary improvement in the analysis is to ensure that $\pi$ does a better job predicting (in a single step) than

$f_t$ does. An obvious open question is whether it is possible to base the analysis on the *regret* of $\pi$ (rather than its error) and whether it is possible to extend beyond the BANDITRON.

# 5 EXPERIMENTAL SETUP AND RESULTS

Using a collection of synthetically generated classification problems, we train an exploration policy $\pi$ using MÊLÉE (Alg 1). This exploration policy learns to explore on the basis of calibrated probabilistic predictions from $f$ together with a predefined set of exploration features (§5.1). Once $\pi$ is learned and fixed, we follow the test-time behavior described in §3.1 to evaluate $\pi$ on a set of contextual bandit problems. We evaluate MÊLÉE on a natural learning to rank task (§5.3.1). To ensure that the performance of MÊLÉE generalizes beyond this single learning to rank task, we additionally perform thorough evaluation on 300 "simulated" contextual bandit problems, derived from standard classification tasks (§5.3.2).

In all cases, the underlying classifier $f$ is a linear model trained with a policy optimizer that runs stochastic gradient descent (details are in §A.2). We seek to answer two questions experimentally: (1) How does MÊLÉE compare empirically to alternative (expert designed) exploration strategies? (2) How important are the additional features used by MÊLÉE in comparison to using calibrated probability predictions from $f$ as features?

## 5.1 TRAINING DETAILS FOR THE EXPLORATION POLICY

**Exploration Features.** In our experiments, the exploration policy is trained based on features $\Phi$ (Alg 1, line 11). These features are allowed to depend on the current classifier $f_t$, and on any part of the history *except* the inputs $x_t$ in order to maintain task independence. We additionally ensure that its features are independent of the *dimensionality* of the inputs, so that $\pi$ can generalize to datasets of arbitrary dimensions. The specific features we use are listed below; these are largely inspired by Konyushkova et al. (2017) but adapted and augmented to our setting.

The **features of** $f_t$ that we use are: **a)** predicted probability $p(a_t|f_t, \boldsymbol{x}_t)$, we use a softmax over the predicted rewards from $f_t$ to convert them to probabilities; **b)** entropy of the predicted probability distribution; **c)** a one-hot encoding for the predicted action $f_t(\boldsymbol{x}_t)$. The **features of** $h_{t-1}$ that we use are: **a)** current time step $t$; **b)** normalized counts for all previous actions predicted so far; **c)** average observed rewards for each action; **d)** empirical variance of the observed rewards for each action in the history. In our experiments, we found that it is essential to calibrate the predicted probabilities of the classifier $f_t$. We use a very small held-out dataset, of size 30, to achieve this. We use Platt's scaling (Platt, 1999; Lin et al., 2007) method to calibrate the predicted probabilities. Platt's scaling works by fitting a logistic regression model to the classifier's predicted scores.

**Training Datasets.** In our experiments, we follow Konyushkova et al. (2017) (and also Peters et al. (2014), in a different setting) and train the exploration policy $\pi$ only on *synthetic data*. This is possible because the exploration policy $\pi$ never makes use of $x$ explicitly and instead only accesses it via $f_t$'s behavior on it. We generate datasets with uniformly distributed class conditional distributions. The datasets are always two-dimensional. Details are in §A.1.

## 5.2 EVALUATION METHODOLOGY

For evaluation, we use progressive validation (Blum et al., 1999), which is exactly computing the reward of the algorithm. Specifically, to evaluate the performance of an exploration algorithm $\mathcal{A}$ on a dataset $S$ of size $n$, we compute the progressive validation return $G(\mathcal{A}) = \frac{1}{n}\sum_{t=1}^{n} r_t(a_t)$ as the average reward up to $n$, where $a_t$ is the action chosen by the algorithm $\mathcal{A}$ and $r_t$ is the true reward.

Because our evaluation is over 300 datasets, we report aggregate results in two forms. The simpler one is **Win/Loss Statistics:** We compare two exploration methods on a given dataset by counting the number of statistically significant wins and losses. An exploration algorithm $\mathcal{A}$ wins over another algorithm $\mathcal{B}$ if the progressive validation return $G(\mathcal{A})$ is statistically significantly larger than $B$'s return $G(\mathcal{B})$ at the 0.01 level using a paired sample t-test. We also report **cumulative distributions** of rewards for each algorithm, following Zhang et al. (2019). In particular, for a given relative reward value ($x \in [0, 1]$), the corresponding CDF value for a given algorithm is the fraction of datasets on which this algorithm achieved reward at least $x$. We compute relative reward by Min-

Max normalization: linearly transforming reward $y$ to $y' = \frac{y - \min}{\max - \min}$, where min & max are the minimum & maximum rewards among all exploration algorithms.

In our experiments, we compare to the following baseline exploration methods, keeping the policy optimization method fixed (details in §A.4): $\epsilon$**-greedy** (Sutton, 1996); $\epsilon$**-decreasing** (Sutton & Barto, 1998); **EG** $\epsilon$**-greedy** (Li et al., 2010b); $\tau$**-first**. We additionally compare to three state-of-the-art exploration methods: **LinUCB** (Li et al., 2010a); **Cover** (Agarwal et al., 2014); and its variant **Cover-NU** (Bietti et al., 2018). We select the best hyperparameters following Bietti et al. (2018).

## 5.3 Experimental Results

### 5.3.1 Learning to Rank

We evaluate MÊLÉE on a natural learning to rank dataset. The dataset we consider is the Microsoft Learning to Rank dataset, variant MSLR-10K from Qin & Liu (2013) [3]. The dataset consists of feature vectors extracted from query-url pairs along with relevance judgment labels. The relevance judgments are obtained from a retired labeling set of a commercial web search engine (Microsoft Bing), which take $5$ values from $0$ (irrelevant) to $4$ (perfectly relevant) and we drop the queries not labelled as any of the two extremes. In our experiments, we limit the number of labels to the two extremes: $0$ and $4$. A query-url pair is represented by a 136-dimensional feature vector. The dataset is highly imbalanced as the number of irrelevant queries is much larger than the number of relevant ones. To address this, we sample the number of irrelevant queries to match that of the relevant ones. To avoid correlations between the observed query-url pairs, we group the queries by the query ID, and sample a single query from each group. We convert relevance scores to losses with 0 indicating a perfectly relevant document, and 1 an irrelevant one.

Figure 1 shows the evaluation results on a subset of the MSLR-10K dataset. Since the performance is closely matched between the different exploration algorithms, we repeat the experiment 16 times with randomly shuffled permutations of the MSLR-10K dataset. Figure 1 (left) shows the learning curve of the trained policy $\pi$ as well as the baselines. Here, we see that MÊLÉE quickly achieves high reward, after about $100$ examples the two strongest baselines catch up. By $200$ examples all approaches have asymptoted. We exclude LinUCB from these runs because the required matrix inversions made it too computationally expensive.[4] Figure 1 shows statistically-significant win/loss differences for each of the algorithms, across these 16 shuffles. Each row/column entry shows the number of times the row algorithm won against the column, minus the number of losses. MÊLÉE is the only algorithm that always wins more than it loses against other algorithms, and outperforms the nearest competition ($\epsilon$-decreasing) by 3 points.

### 5.3.2 Simulated Contextual Bandit Tasks

We additionally perform an exhaustive evaluation on simulated contextual bandit tasks to ensure that the performance of MÊLÉE generalizes beyond learning to rank. Following Bietti et al. (2018), we use a collection of 300 binary classification datasets from `openml.org` for evaluation; the precise list and download instructions is in §A.3. These datasets cover a variety of different domains including text & image processing, medical, and sensory data. We convert classification datasets into cost-sensitive classification problems by using a $0/1$ encoding. Given these fully supervised cost-sensitive multi-class datasets, we simulate the contextual bandit setting by only revealing the reward for the selected actions.

In Figure 2 (left), we show a representative learning curve. Here, we see that as more data becomes available, all the approaches improve (except $\tau$-first, which has ceased to learn after $2\%$ of the data). MÊLÉE, in particular, is able to very quickly achieve near-optimal performance (in around $40$ examples) in comparison to the best baseline which takes at least $200$. In Figure 2 (right), we show the CDFs for the different algorithms. To help read this, at $x = 1.0$, MÊLÉE has a relative reward at least 1.0 on more than 40% of datasets, while $\epsilon$-decreasing and $\epsilon$-greedy achieve this on about 30% of datasets. We find that the two strongest baselines are $\epsilon$-decreasing and $\epsilon$-greedy (better when reward differences are small, toward the left of the graph). The two curves for $\epsilon$-decreasing and $\epsilon$-greedy coincide. This happens because the exploration probability $\epsilon_0$ for $\epsilon$-decreasing decays rapidly

---

[3] `https://www.microsoft.com/en-us/research/project/mslr/`

[4] In a single run of LinUCB we observed that its performance is on par with $\epsilon$-greedy.

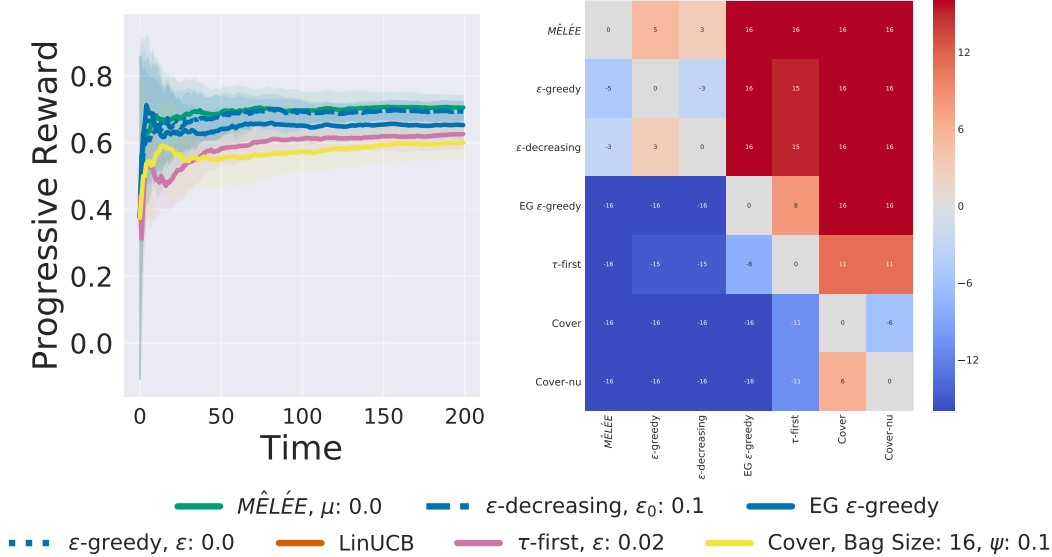

Figure 1: Results for the Learning to Rank task. **(Left)** Learning curve on the MSLR-10K dataset: x-axis shows the number of queries observed, and y-axis shows the progressive reward. **(Right)** Win/Loss counts for all pairs of algorithms over 16 random shuffles for the MSLR-10K dataset.

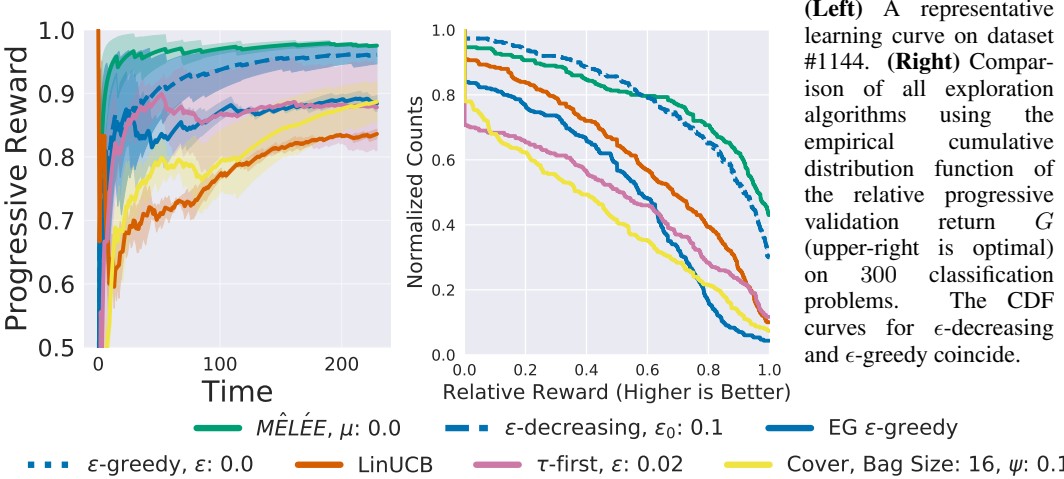

Figure 2: Behavior of MÊLÉE in comparison to baseline and state-of-the-art exploration algorithms.

approaching zero with a rate of $\frac{1}{t}$, where $t$ is the index of the current round. MÊLÉE outperforms the baselines in the "large reward" regimes (right of graph) but under-performs $\epsilon$-decreasing and $\epsilon$-greedy in low reward regimes (left of graph).

In Figure 3, we show statistically-significant win/loss differences for each of the algorithms. Here, each (row, column) entry shows the number of times the row algorithm won against the column, minus the number of losses. MÊLÉE is the only algorithm that always wins more than it loses against other algorithms, and outperforms the nearest competition ($\epsilon$-decreasing) by 23 points.

To understand more directly how MÊLÉE compares to $\epsilon$-decreasing, in the left plot of Figure 4, we show a scatter plot of rewards achieved by MÊLÉE (x-axis) and $\epsilon$-decreasing (y-axis) on each of the 300 datasets, with statistically significant differences highlighted in red and insignificant differences in blue. Points below the diagonal line correspond to better performance by MÊLÉE (147 datasets)

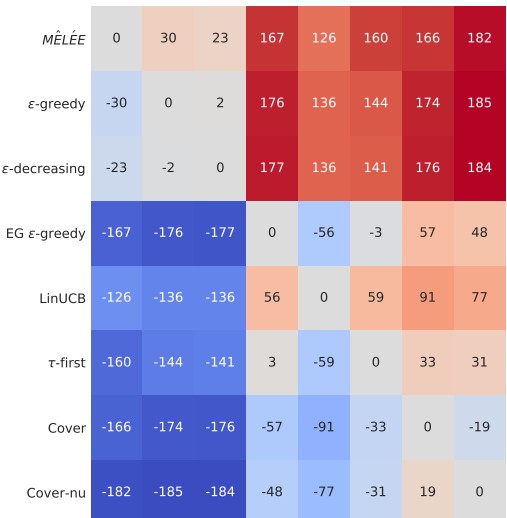

Figure 3: Win/Loss counts for all pairs of algorithms (columns match the rows).

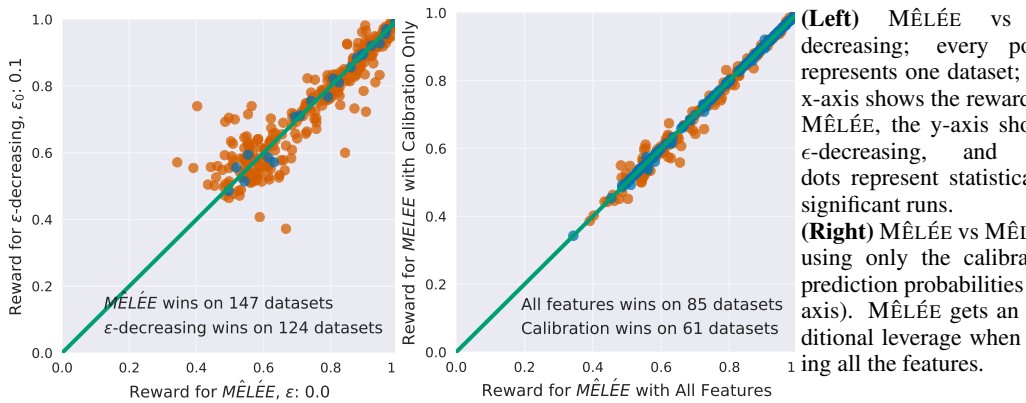

**(Left)** MÊLÉE vs $\epsilon$-decreasing; every point represents one dataset; the x-axis shows the reward of MÊLÉE, the y-axis shows $\epsilon$-decreasing, and red dots represent statistically significant runs.
**(Right)** MÊLÉE vs MÊLÉE using only the calibrated prediction probabilities (x-axis). MÊLÉE gets an additional leverage when using all the features.

Figure 4: Scatterplots comparing MÊLÉE to the best baseline and to a variant with fewer features.

and points above to $\epsilon$-decreasing (124 datasets). The remaining 29 had no statistically significant difference.

Finally, we consider the effect that the additional features have on MÊLÉE's performance. In particular, we consider a version of MÊLÉE with all features (this is the version used in all other experiments) with an ablated version that only has access to the (calibrated) probabilities of each action from the underlying classifier $f$. The comparison is shown as a scatter plot in Figure 4 (right). Here, we can see that the full feature set *does* provide lift over just the calibrated probabilities, with a win-minus-loss improvement of 24 by adding additional features from which to learn to explore.

## 6 RELATED WORK AND DISCUSSION

The field of meta-learning is based on the idea of replacing hand-engineered learning heuristics with heuristics learned from data. One of the most relevant settings for meta-learning to ours is active learning, in which one aims to learn a decision function to decide which examples, from a pool of unlabeled examples, should be labeled. Past approaches to meta-learning for active learning include reinforcement learning-based strategies (Woodward & Finn, 2017; Fang et al., 2017), imitation learning-based strategies (Bachman et al., 2017), and batch supervised learning-based strategies (Konyushkova et al., 2017). Similar approaches have been used to learn heuristics for optimization (Li & Malik, 2016; Andrychowicz et al., 2016), multiarm (non-contextual) bandits Maes et al. (2012), and neural architecture search (Zoph & Le, 2016), recently mostly based on (deep) reinforcement learning. While meta-learning for contextual bandits is most similar to meta-learning for

active learning, there is a fundamental difference that makes it significantly more challenging: in active learning, the goal is to select as few examples as you can to learn, so by definition the horizon is short; in contextual bandits, learning to explore is fundamentally a long-horizon problem, because what matters is not immediate reward but long term learning.

In reinforcement learning, Gupta et al. (2018) investigated the task of meta-learning an exploration strategy for a distribution of related tasks by learning a latent exploration space. Similarly, Xu et al. (2018) proposed a teacher-student approach for learning to do exploration in off-policy reinforcement learning. While these approaches are effective if the distribution of tasks is very similar and the state space is shared among different tasks, they fail to generalize when the tasks are different. Our approach targets an easier problem than exploration in full reinforcement learning environments, and can generalize well across a wide range of different tasks with completely unrelated features spaces.

There has also been a substantial amount of work on constructing "good" exploration policies, in problems of varying complexity: traditional bandit settings (Karnin & Anava, 2016), contextual bandits (Fraud et al., 2016) and reinforcement learning (Osband et al., 2016). In both bandit settings, most of this work has focused on the learning theory aspect of exploration: what exploration distributions *guarantee* that learning will succeed (with high probability)? MÊLÉE, lacks such guarantees: in particular, if the data distribution of the observed contexts $(\phi(f_t))$ in some test problem differs substantially from that on which MÊLÉE was trained, we can say nothing about the quality of the learned exploration. Nevertheless, despite fairly substantial distribution mismatch (synthetic $\rightarrow$ real-world), MÊLÉE works well in practice, and our stylized theory (§4) suggests that there may be an interesting avenue for developing strong theoretical results for contextual bandit learning with learned exploration policies, and perhaps other meta-learning problems.

In conclusion, we presented MÊLÉE, a meta-learning algorithm for learning exploration policies in the contextual bandit setting. MÊLÉE enjoys no-regret guarantees, and empirically it outperforms alternative exploration algorithm in most settings. One limitation of MÊLÉE is the computational resources required during the offline training phase on the synthetic datasets. In the future, we will work on improving the computational efficiency for MÊLÉE in the offline training phase and scale the experimental analysis to problems with larger number of classes.

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

# Supplementary Material For:
# Learning Effective Exploration Strategies for Contextual Bandits

## A  STYLIZED TEST-TIME ANALYSIS FOR BANDITRON: DETAILS

The BANDITRONMÊLÉE algorithm is specified in Alg 2. The is exactly the same as the typical test time behavior, except it uses a BANDITRON-type strategy for learning the underlying classifier $f$ in the place of POLOPT. POLICYELIMINATIONMETA takes as arguments: $\pi$ (the learned exploration policy) and $\mu \in (0, 1/(2K))$ an added uniform exploration parameter. The BANDITRON learns a linear multi-class classifier parameterized by a weight matrix of size $K \times D$, where $D$ is the input dimensionality. The BANDITRON assumes a pure multi-class setting in which the reward for one ("correct") action is 1 and the reward for all other actions is zero.

At each round $t$, a prediction $\hat{a}_t$ is made according to $f_t$ (summarized by $W^t$). We then define an exploration distribution that "most of the time" acts according to $\pi(f_t, .)$, but smooths each action with $\mu$ probability. The chosen action $a_t$ is sampled from this distribution and a binary reward is observed. The weights of the BANDITRON are updated according to the BANDITRON update rule using $\tilde{U}^t$.

---

**Algorithm 2** BANDITRONMÊLÉE $(g, \mu)$

---

1: initialize $W^1 = \mathbf{0} \in \mathbb{R}^{K \times D}$
2: **for** rounds $t = 1 \ldots T$: **do**
3:     observe $x_t \in \mathbb{R}^D$
4:     compute $\hat{a}_t = f_t(x_t) = \text{argmax}_{k \in K} \left(W^t x_t\right)_k$
5:     define $Q^\mu(a) = \mu + (1 - K\mu)\mathbf{1}[a = \pi(W^t, x_t)]$
6:     sample $a_t \sim Q^\mu$
7:     observe reward $r_t(a_t) \in \{0, 1\}$
8:     define $\tilde{U}^t \in \mathbb{R}^{K \times D}$ as:
        $\tilde{U}^t_{a,\cdot} = x_t \left( \frac{\mathbf{1}[r_t(a_t)=1]\mathbf{1}[a_t=a]}{Q^\mu(a)} - \mathbf{1}[\hat{a}_t = a] \right)$
9:     update $W^{t+1} = W^t + \tilde{U}^t$
10: **end for**

---

The *only* difference between BANDITRONMÊLÉE and the original BANDITRON is the introduction of $\pi$ in the sampling distribution. The original algorithm achieves the following mistake bound shown below, which depends on the notion of multi-class hinge-loss. In particular, the hinge-loss of $W$ on $(x, \boldsymbol{r})$ is $\ell(W, (x, \boldsymbol{r})) = \max_{a \neq a^\star} \max \left\{0, 1 - (Wx)_{a^\star} + (Wx)_a\right\}$, where $a^\star$ is the $a$ for which $r(a) = 1$. The overall hinge-loss $L$ is the sum of $\ell$ over the sequence of examples.

**Theorem 3 (Thm. 1 and Corr. 2 of Kakade et al. (2008))** *Assume that for the sequence of examples, $(x_1, \boldsymbol{r}_1), (x_2, \boldsymbol{r}_2), \ldots, (x_T, \boldsymbol{r}_T)$, we have, for all $t$, $||x_t|| \leq 1$. Let $W^\star$ be any matrix, let $L$ be the cumulative hinge-loss of $W^\star$, and let $D = 2 ||W^\star||_F^2$ be the complexity of $W^\star$. The number of mistakes $M$ made by the BANDITRON satisfies*

$$\mathbb{E}M \leq L + K\mu T + 3 \max \left\{ \frac{D}{\mu}, \sqrt{DTK\mu} \right\} + \sqrt{DL/\mu} \tag{4}$$

*where the expectation is taken with respect to the randomness of the algorithm. Furthermore, in a low noise setting (there exists $W^\star$ with fixed complexity $d$ and loss $L \leq O(\sqrt{DKT})$), then by setting $\mu = \sqrt{D/(TK)}$, we obtain $\mathbb{E}M \leq O(\sqrt{KDT})$.*

We can prove an analogous result for BANDITRONMÊLÉE. The key quantity that will control how much $\pi$ improves the execution of BANDITRONMÊLÉE is how much $\pi$ improves on $f_t$ when $f_t$ is wrong. In particular, let $\gamma_t = \Pr[r_t(\pi(f_t, x_t)) = 1)|x_t] - \Pr[r_t(f_t(x_t)) = 1|x_t]$ be the edge of $\pi(f_t, .)$ over $f$, and let $\Gamma = \frac{1}{T} \sum_{t=1}^T \mathbb{E} \frac{1}{1+K\gamma_t}$ be an overall measure of the edge. (If $\pi$ does nothing, then all $\gamma_t = 0$ and $\Gamma = 1$.) Given this quantity, we can prove the following Theorem 2.

**Proof:** [sketch] The proof is a small modification of the original proof of Theorem 3. The only change is that in the original proof, the following bound is used: $\mathbb{E}_t ||\tilde{U}^t||^2 / ||x_t||^2 = 1 + 1/\mu \leq 2/\mu$.

We use, instead: $\mathbb{E}_t||\tilde{U}^t||^2/||x_t||^2 \leq 1 + \mathbb{E}_t\frac{1}{\mu+\gamma_t} \leq \frac{2\mathbb{E}_t\frac{1}{1+\gamma_t}}{\mu}$. The rest of the proof goes through identically. □

## A.1 DETAILS OF SYNTHETIC DATASETS

We generate datasets with uniformly distributed class conditional distributions. We generate 2D datasets by first sampling a random variable representing the Bayes classification error. The Bayes error is sampled uniformly from the interval $0.0$ to $0.5$. Next, we generate a balanced dataset where the data for each class lies within a unit rectangle and sampled uniformly. We overlap the sampling rectangular regions to generate a dataset with the desired Bayes error selected in the first step.

## A.2 IMPLEMENTATION DETAILS

Our implementation is based on scikit-learn (Pedregosa et al., 2011). We fix the training time exploration parameter $\mu$ to $0.1$. We train the exploration policy $\pi$ on 82 synthetic datasets each of size $3000$ with uniform class conditional distributions, a total of $246k$ samples (§A.1). We train $\pi$ using a linear classifier Breiman (2001) and set the hyper-parameters for the learning rate, and data scaling methods using three-fold cross-validation on the whole meta-training dataset. For the classifier class $\mathcal{F}$, we use a linear model trained with stochastic gradient descent. We standardize all features to zero mean and unit variance, or scale the features to lie between zero and one. To select between the two scaling methods, and tune the classifier's learning rate, we use three-fold cross-validation on a small fully supervised training set of size 30 samples. The same set is used to calibrate the predicted probabilities of $f_t$.

## A.3 LIST OF DATASETS

The datasets we used can be accessed at `https://www.openml.org/d/<id>`. The list of $(\text{id}, \text{size})$ pairs below shows the (id for the datasets we used and the dataset size in number of examples:

(46,100) (716, 100) (726, 100) (754, 100) (762, 100) (768, 100) (775, 100) (783, 100) (789, 100) (808, 100) (812, 100) (828, 100) (829, 100) (850, 100) (865, 100) (868, 100) (875, 100) (876, 100) (878, 100) (916, 100) (922, 100) (932, 100) (1473, 100) (965, 101) (1064, 101) (956, 106) (1061, 107) (771, 108) (736, 111) (448, 120) (782, 120) (1455, 120) (1059, 121) (1441, 123) (714, 125) (867, 130) (924, 130) (1075, 130) (1141, 130) (885, 131) (444, 132) (921, 132) (974, 132) (719, 137) (1013, 138) (1151, 138) (784, 140) (1045, 145) (1066, 145) (1125, 146) (902, 147) (1006, 148) (969, 150) (955, 151) (1026, 155) (745, 159) (756, 159) (1085, 159) (1054, 161) (748, 163) (747, 167) (973, 178) (463, 180) (801, 185) (1164, 185) (788, 186) (1154, 187) (941, 189) (1131, 193) (753, 194) (1012, 194) (1155, 195) (1488, 195) (446, 200) (721, 200) (1124, 201) (1132, 203) (40, 208) (733, 209) (796, 209) (996, 214) (1005, 214) (895, 222) (1412, 226) (820, 235) (851, 240) (464, 250) (730, 250) (732, 250) (744, 250) (746, 250) (763, 250) (769, 250) (773, 250) (776, 250) (793, 250) (794, 250) (830, 250) (832, 250) (834, 250) (863, 250) (873, 250) (877, 250) (911, 250) (918, 250) (933, 250) (935, 250) (1136, 250) (778, 252) (1442, 253) (1449, 253) (1159, 259) (450, 264) (811, 264) (336, 267) (1152, 267) (53, 270) (1073, 274) (1156, 275) (880, 284) (1121, 294) (43, 306) (818, 310) (915, 315) (1157, 321) (1162, 322) (925, 323) (1140, 324) (1144, 329) (1011, 336) (1147, 337) (1133, 347) (337, 349) (59, 351) (1135, 355) (1143, 363) (1048, 369) (860, 380) (1129, 384) (1163, 386) (900, 400) (906, 400) (907, 400) (908, 400) (909, 400) (1025, 400) (1071, 403) (1123, 405) (1160, 410) (1126, 412) (1122, 413) (1127, 421) (764, 450) (1065, 458) (1149, 458) (1498, 462) (724, 468) (814, 468) (1148, 468) (1150, 470) (765, 475) (767, 475) (1153, 484) (742, 500) (749, 500) (750, 500) (766, 500) (779, 500) (792, 500) (805, 500) (824, 500) (838, 500) (855, 500) (869, 500) (870, 500) (879, 500) (884, 500) (886, 500) (888, 500) (896, 500) (920, 500) (926, 500) (936, 500) (937, 500) (943, 500) (987, 500) (1470, 500) (825, 506) (853, 506) (872, 506) (717, 508) (1063, 522) (954, 531) (1467, 540) (1165, 542) (1137, 546) (335, 554) (333, 556) (947, 559) (949, 559) (950, 559) (951, 559) (826, 576) (1004, 600) (334, 601) (1158, 604) (770, 625) (997, 625) (1145, 630) (1443, 661) (774, 662) (795, 662) (827, 662) (931, 662) (292, 690) (1451, 705) (1464, 748) (37, 768) (1014, 797) (970, 841) (994, 846) (841, 950) (50, 958) (1016, 990) (31, 1000) (715, 1000) (718, 1000) (723, 1000) (740, 1000) (743, 1000) (751, 1000) (797, 1000) (799, 1000) (806, 1000) (813, 1000) (837, 1000) (845, 1000) (849, 1000) (866, 1000) (903, 1000) (904, 1000) (910, 1000) (912, 1000) (913, 1000) (917, 1000) (741, 1024) (1444, 1043) (1453, 1077)

(1068, 1109) (934, 1156) (1049, 1458) (1454, 1458) (983, 1473) (1128, 1545) (1130, 1545) (1138, 1545) (1139, 1545) (1142, 1545) (1146, 1545) (1161, 1545) (1166, 1545) (1050, 1563) (991, 1728) (962, 2000) (971, 2000) (978, 2000) (995, 2000) (1020, 2000) (1022, 2000) (914, 2001) (1067, 2109) (772, 2178) (948, 2178) (958, 2310) (312, 2407) (1487, 2534) (737, 3107) (953, 3190) (3, 3196) (1038, 3468) (871, 3848) (728, 4052) (720, 4177) (1043, 4562) (44, 4601) (979, 5000) (1460, 5300) (1489, 5404) (1021, 5473) (1069, 5589) (980, 5620) (847, 6574) (1116, 6598) (803, 7129) (1496, 7400) (725, 8192) (735, 8192) (752, 8192) (761, 8192) (807, 8192)

## A.4 Baseline Exploration Algorithms

Our experiments aim to determine how MÊLÉE compares to other standard exploration strategies. In particular, we compare to:

$\epsilon$**-greedy:** With probability $\epsilon$, explore uniformly at random; with probability $1 - \epsilon$ act greedily according to $f_t$ (Sutton, 1996). Experimentally, we found $\epsilon = 0$ optimal on average, consistent with the results of Bietti et al. (2018).

$\epsilon$**-decreasing:** selects a random action with probabilities $\epsilon_i$, where $\epsilon_i = \epsilon_0/t$, $\epsilon_0 \in ]0, 1]$ and $t$ is the index of the current round. In our experiments we set $\epsilon_0 = 0.1$. (Sutton & Barto, 1998)

**Exponentiated Gradient $\epsilon$-greedy:** maintains a set of candidate values for $\epsilon$-greedy exploration. At each iteration, it runs a sampling procedure to select a new $\epsilon$ from a finite set of candidates. The probabilities associated with the candidates are initialized uniformly and updated with the Exponentiated Gradient (EG) algorithm. Following Li et al. (2010b), we use the candidate set $\{\epsilon_i = 0.05 \times i + 0.01, i = 1, \cdots, 10\}$ for $\epsilon$.

**LinUCB:** Maintains confidence bounds for reward payoffs and selects actions with the highest confidence bound. It is impractical to run "as is" due to high-dimensional matrix inversions. We use diagonal approximation to the covariance when the dimensions exceeds 150. (Li et al., 2010a)

$\tau$**-first:** Explore uniformly on the first $\tau$ fraction of the data; after that, act greedily.

**Cover:** Maintains a uniform distribution over a fixed number of policies. The policies are used to approximate a covering distribution over policies that are good for both exploration and exploitation (Agarwal et al., 2014).

**Cover Non-Uniform:** similar to Cover, but reduces the level of exploration of Cover to be more competitive with the Greedy method. Cover-Nu doesn't add extra exploration beyond the actions chose by the covering policies (Bietti et al., 2018).

In all cases, we select the best hyperparameters for each exploration algorithm following Bietti et al. (2018). These hyperparameters are: the choice of $\epsilon$ in $\epsilon$-greedy, $\tau$ in $\tau$-first, the number of bags, and the tolerance $\psi$ for Cover and Cover-NU. We set $\epsilon = 0.0$, $\tau = 0.02$, bag size $= 16$, and $\psi = 0.1$.

