# OpenReview forum: "Learning Effective Exploration Strategies For Contextual Bandits"
_ICLR.cc/2020/Conference — Reject_

### Official Review · AnonReviewer1 · 2019-10-18
**Official Blind Review #1**

**Rating:** 1

**Review:**

This paper presents an approach to learning exploration strategies for contextual bandits by meta-learning the exploration strategy across synthetic tasks. The approach is shown to outperform alternative exploration strategies on a learning to rank task as well as simulated bandit problems derived from classification tasks.

Applying meta-learning to this setting is a novel and interesting approach and appears to have nice results.

The main issue with this paper is that the presentation of the algorithm is vague and it is difficult to determine what the algorithm actually does. Here are a few of my main questions:

1. The algorithm is dependent on a function class F and PolOpt which finds an f in F with low expected regret. However, F is not really defined. In equation 1, it appears to directly output an action. Below that, it says "For example, F may be... mapping user features x in X to predicted rewards for actoins a in [K]." In Section 5.1, the policy is using a probability distribution over actions from f, the entropy of the predicted probability distribution, and a one hot encoding for the predicted action f(a). What is the class F? Is it predicting a probability distribution and an action? What's the one hot encoding of the predicted action, is that an action sampled from the probability distribution of F?

2. Section 3.1 describes the test time behavior of MELEE. They state the goal is to learn an exploration policy pi, and specifically state that pi can "not depend directly on the context x" so it "only learns to explore and not also to solve the underlying task-dependent classification problem". However, in the equations, pi is a function of x_t. The algorithm is motivated by the exploration exploitation trade-off problem, however there's no policy doing any exploiting here.  When does exploitation happen? Is it a separate policy? How is it trained and when is it used? The decision of when to explore and when to exploit is the key question here and it's unclear how that is happening in this algorithm. It's unclear to me how the algorithm achieves good rewards on the task without any attempt at exploitation.

3. What are the rollouts and expert cost-to-go in this setting? In section 3.2, you say the algorithm asks what would the "long time reward look like if I were to take this action." So are you looking at how that action would affect future exploitation? You say there's an optimal reference policy* which effectively cheats by looking at the true labels. Is this policy taking the action with the highest reward at each state? Or is it looking at the effects of exploration on the learned policy? The start of page 4 says that the "cost-to-go Q of the expert" is observed, why would the expert in a bandit have a cost-to-go? In the section on roll-out values, you say that you evaluate the value of taking this action and then assuming all future actions were taken by the expert. But this then precludes there being any effect of this action on the policy and any exploration effect. Are you solely optimizing for the one step reward? In the algorithm, the rollouts computed on line 8 don't appear to be used anywhere.

4. How are the synthetic tasks setup? There are almost no details other than that they are two dimensional and have uniformly distributed class conditional distributions. How are they related to the learning to rank task you perform at the end? How are they related to the other tasks you evaluate on? It seems like the choice of synthetic tasks would be important to what the algorithm learns, so it's important to be clear on what these tasks were and how closely they match the target tasks. It would be even better to show results with different sets of synthetic tasks that are more or less similar to the target tasks.

5. In the results, you say you limit the number of labels to two extremes, 0 and 4. I assume you mean you limit it to just those two labels. What do you do with the other data? Do you drop it or do you map it to these two labels?

**Experience Assessment:**

I have read many papers in this area.

**Review Assessment: Checking Correctness Of Derivations And Theory:**

I did not assess the derivations or theory.

**Review Assessment: Checking Correctness Of Experiments:**

I carefully checked the experiments.

**Review Assessment: Thoroughness In Paper Reading:**

I read the paper thoroughly.

---

> ### Author Response · Authors · 2019-11-12
> **Response to official blind review #1**
>
> We thank the reviewer for the provided feedback. We clarify several points below, specially relating to how exploitation is handled in our algorithm. As the reviewer highlighted, the exploitation / exploration tradeoff is the key question studied in our paper.
>
> To address a high-level question first, in question 2, the reviewer asks about "when does exploitation happen?" There is no separate exploitation policy: exploitation happens when \pi chooses the same action as f, while exploration happens when it chooses a different action. In learning to explore, we are simultaneously learning when to exploit.
>
> We attempted in the paper to be precise, and we thank the reviewer for raising questions where things were unclear. If there are additional points that seem vague, we would appreciate the opportunity to address them.
>
> On the specific questions:
>
> 1) There are two questions here:
>
> a) Function class F: we apologize for any confusion here. In the discussion around Eq (1) we were attempting not to commit precisely to the form of F, but this perhaps led to more confusion. As stated below Eq (1), F is the class of functions from input features to predicted rewards for each action, and in Eq (1), "f(x_t)" is shorthand for "pick the action with maximal predicted reward." POLOPT is optimizing over that class. In the experiment section, where we use per-class probabilities, we use a softmax over the predicted rewards to convert them to probabilities. (We forgot to add the last point to the paper and will clarify both of these issues.)
>
> b) One hot encoding of the predicted action: the classifier f predicts an action for the given feature vector x. The one hot encoding is just an encoding for this predicted action.
>
> 2) This piece has two questions:
>
> a) \pi depending on x: The features we use (see sec 5.1) do not depend on x directly, but instead on f's behavior on x. Perhaps it would be more clear to write this as \pi(f(x)) rather than \pi(f,x). The meta-features we used have no direct dependence on the context vector x. We include however P(a|x) as a meta-feature, where P(a|x) measures the confidence of the classifier f on the context vector x.
>
> b) When does exploitation happen: see above.
>
> 3) This has several subquestions:
>
> a) Roll-outs: When we assume all future decisions are taken optimally, the optimal behavior at time t becomes simply choosing the action with the immediate highest reward (i.e., as the reviewer stated, the roll-out is not looking at how that action would affect future exploration.)
>
> b) Cost-to-go: this is cost-to-go in the *imitation* problem, not in the bandit problem
>
> c) Roll-outs computed: they are used in line 11 as the target of the classifier
>
> 4) Synthetic datasets: we generated synthetic datasets for training and experimented with different strategies for generating these datasets. The simple approach described in A.1 worked best, as it controls for the difficulty of the learning task. In practice, these are very dissimilar to any of the actual settings (either the L2R setting or the simulated CB setting), yet are sufficient to learn a good exploration policy (due to the lack of explicit dependence on x).
>
> 5) Limiting the ranking data: we drop the queries not labelled as any of the two extremes.

---

> > ### Comment · AnonReviewer1 · 2019-11-14
> > **Response**
> >
> > Thanks for the feedback and answering the questions. Once you can make all of these clearer in the paper, it will make it much stronger.

---

### Official Review · AnonReviewer3 · 2019-10-23
**Official Blind Review #3**

**Rating:** 1

**Review:**

This paper proposes a meta-learning algorithm to solve the problem of exploration in a contextual bandit task using prior knowledge. This is analogous to how exploration strategies have been learned from past data in the meta-learning for RL (for example, [1]). Their algorithm simulates contextual bandit problems from fully labeled data and uses it to learn an exploration policy that works well on tasks with bandit feedback. The training step for the exploration policy builds upon the AggreVaTe algorithm, where policy optimization is performed on the history augmented data by using a separate roll-out policy to estimate the advantage of a particular action for a particular context, from the point of view of regret minimization.  In terms of theory, they show that by using specific algorithms (example, Banditron) as the inner policy optimization procedure, a no-regret algorithm can be obtained.

Some of my questions are:

1. The method seems a bit hacky to me, for example, it requires calibration of f, requires access to test time examples, and it is unclear why this should be needed if an algorithm were provably good. Can this be elaborated upon?

2. Section 3.2 and Algorithm 1 are very unclear, and it requires multiple passes to be able to understand in the current draft. I encourage the authors to revise these sections for increasing clarity.

3. In Algorithm 1, which is the exploitation policy? The theory section says " In particular, we first relate the
regret of the learner in line 16 to the overall regret of $pi$", but line 16 in the algorithm refers to "end for". Can the authors clearly point me to the location where the exploitation policy is mentioned in the algorithm? What is the precise definition of $\pi_n$ (Line 15), and how does it relate to the average exploration policy and the policy $\bar{\pi}$ used in Theorem 1. These definitions have not been made clear, making this hard to follow. What is the optimal policy $\pi^*$, in particular, what is the formal meaning of "which effectively cheats” at training time by looking at the true labels."

4. The paper lacks discussion about how the roll-out and the roll-in policy affect regret bounds, and what assumption about these is used to derive no-regret guarantees. Can this be described?

Overall, my recommendation currently is reject. I feel that the paper tries to theoretically analyse an interesting problem, however, at this point the paper is very hard to follow and not complete. I encourage the authors to revise the details and improve clarity. Also, it would be good if the authors can explain the significance of the results at the cost of added complexity.

References:
[1] Meta-exploration of structured exploration strategies, Gupta et.al. NeuRIPS 2018

**Experience Assessment:**

I have read many papers in this area.

**Review Assessment: Checking Correctness Of Derivations And Theory:**

I assessed the sensibility of the derivations and theory.

**Review Assessment: Checking Correctness Of Experiments:**

I assessed the sensibility of the experiments.

**Review Assessment: Thoroughness In Paper Reading:**

I read the paper at least twice and used my best judgement in assessing the paper.

---

> ### Author Response · Authors · 2019-11-12
> **Response to Official Blind Review #3:**
>
> We thank the reviewer for the provided feedback. We address the reviewer’s questions below.
>
> 1) Calibration of f and access to test time examples: we agree that removing the dependence on the small number of test examples for calibration would be preferable. Unfortunately, we found that if f’s predictions were uncalibrated, it was very difficult for the exploration policy \pi to generalize well to unseen tasks, and we could not find calibration methods that did not require access to such data. We are very open to suggestions and pointers to alternative ways to achieve calibrated predictions.
>
> 2) Clarity of sec 3.2: we thank the reviewer for highlighting the unclarity in this section. We’ll incorporate the reviewer’s suggestions to make this section more clear.
>
> 3) This question has many subparts.
>
> a) Exploitation policy: there is no explicit "exploitation policy" -- exploitation happens when \pi chooses the same action as f (while exploration happens when it chooses a different action).
>
> b) Line 16: sorry, we're not sure what happened here; the reference should be to line 15 (where \pi is learned)
>
> c) Precise definition of \pi_n: this is the policy learned in the nth round (per line 15)
>
> d) Average policy: the algorithm returns a sequence of policies in line 17 ({\pi_n}_{n=1}^N); the average policy \bar\pi is the average of these returned policies (as stated in the theorem statement)
>
> e) Optimal policy \pi*: in the "Roll-out values" paragraph (p4), we define the rollout policy to be the one that always chooses the action with the highest reward, which is optimal under the "all future actions taken by expert" approach.
>
> 4) Effect of roll-out / roll-in strategy on regret guarantees: The theorem is stated with respect to some fixed policy \pi* and holds for any definition thereof; in our case, \pi* is defined by the rollout, and achieves maximal reward. (Note: we just noticed a typo in the statement of the theorem; the inequality should go the other way, and the + should be a -... this error occurred because of switching from loss-semantics to reward-semantics.) For roll-in, we are always rolling in with the most recent learned policy; if we were using something else, there would be an additional term in the regret bound (making it worse).
>
> We appreciate that the reviewer took the time to understand the paper and if there are any specific suggestions for improving the clarity, we would welcome them.
>
> Significance of the results at the cost of added complexity:
> ==================================================
>
>  First, to clarify the "test time" complexity, the pieces that need to be "shipped" are (a) the calibration and (b) the policy (which is a scikit learn classifier). In comparison to \epsilon-greedy, which takes one line of implementation, this is certainly more complex, but remarkably generalizable given the substantial difference between the synthetic training data and the test settings. It's seems hard to have lower implementation complexity than epsilon-greedy, but in many ways the learned policy here is much simpler than alternatives like cover-nu and UCB.

---

### Official Review · AnonReviewer2 · 2019-10-24
**Official Blind Review #2**

**Rating:** 3

**Review:**

•	Summary
    This paper introduced a meta-learning algorithm for the contextual bandit problem, MELEE, which learns an exploration policy based on simulated and synthetic contextual bandit tasks. The training is mainly divided into two steps. In step one, they proposed to train a policy optimizer, which maps features and actions to rewards. This policy optimizer could be used to reveal the most valuable action to take according to the modeled reward. All possible actions and their corresponding values are revealed to the policy optimizer because of the existing ground-truth labels in the synthetic dataset. The policy optimizer would then suggest which action to take. The algorithm takes the action in an  greedy fashion, i.e. with probability  it will follow the suggestion and with probability  it will sample it uniformly at random. The policy optimizer, historical actions and the taken actions are appended to the training set for training the exploration policy  in the next step. The procedure in step one is proposed to be done in  rounds. In step two, the training set is used for training an exploration policy  . During testing, the contexts are drawn from the real world, the policy optimizer will first evaluate the whole history and the exploration policy will generate actions with the input from the policy optimizer and the context. The algorithm suggests the action to explore in an  greedy fashion. The proposed algorithm is evaluated on a dataset for learning to rank, and 300 synthetic datasets. It shows better performances in most cases.
    I am critical about the paper  because 1) the experiments show that all more recently published exploration methods are worse than weak baselines, for which it lacks enough justification and convincing explanations. 2) the experiments are difficult to understand with only citation to publications. Detailed information of the datasets, tasks, procedures is missing.
•	Main arguments
    The paper is in general hard to follow because of too many citations of the previous works without simple explanations. It refers to the imitation algorithm, AggreVate, which is an instantiate of meta-learning for contextual bandits. Meanwhile, they failed to clarify the difference between the proposed algorithm and the existing one, making the training algorithm part confusing. The major concern lies in the experiment section, I can not see big performance difference between the proposed method and the  greedy based methods in Figure 1 (left) as 1) the variances are large 2) there are overlapping . It is surprising to see all recently published method are worse than the classical  greedy method. These results may require deeper investigations. In addiction, the used datasets in experiments are mentioned without any details and task definitions, which makes the experiments part unclear and the result not that convincing. Below are some other inconsistencies in the paper:
1.	I had a hard time to understand what the function  means. In Section 2,  is used to map user feature to predicted rewards for actions, i.e.  is a function with user features and actions as input and reward as output. However, in equation 1, it only takes user feature as input. In Section 5.1,  is called a classifier (I regard it as a variant of function  ). Both are not consistent with the mapping definition when first formally defined. This point confuses me so that I cannot fully understand what the POLOPT does as the function  is the output of it.
2.	By the end of section 2, the paper claims that they used direct method for it simplicity and unbiased property. However, as verified in [1], the direct method is biased with low variance whereas IPS is unbiased with high variance.
[1] Dudık, Miroslav, John Langford, and Lihong Li. “Doubly Robust Policy Evaluation and Learning.”


•	Things to improve that did not impact the score
a.	clear definition of the introduced notations, including the ones in algorithms 1.
•	Questions:
a.	What is  ?
b.	What is POLOPT?
c.	Why do you choose AggreVate?
d.	Will the performance change if using any other methods instead of direct method for policy optimizer?
e.	What is roll-out policy in line 8 of Algorithm 1? (In text only the roll-out value is defined.)
f.	What will happen if the roll-in action is different from the behavior in test time?
g.	How is exploration policy trained?


**Experience Assessment:**

I have read many papers in this area.

**Review Assessment: Checking Correctness Of Derivations And Theory:**

I did not assess the derivations or theory.

**Review Assessment: Checking Correctness Of Experiments:**

I assessed the sensibility of the experiments.

**Review Assessment: Thoroughness In Paper Reading:**

I read the paper at least twice and used my best judgement in assessing the paper.

---

> ### Author Response · Authors · 2019-11-12
> **Response to Official Blind Review #2 :**
>
> We thank the reviewer for the feedback. We believe there is a fundamental misunderstanding of our work. It’s not true (as the reviewer states) that we propose a two-step training algorithm to solve contextual bandit problems. Our goal, as stated in the title, abstract, introduction, and throughout the paper is: “We develop a meta-learning algorithm, MELEE, that learns an exploration policy based on simulated, synthetic contextual bandit tasks”. We focus solely on learning the exploration policy, which can be used with any contextual bandit policy optimizer ("POLOPT") to solve the contextual bandit problem. As stated in the approach section 3.1 and further in the experiments section 4: the output from MELEE is an exploration policy \pi, which is then fixed and used for exploration at test time as described in 3.1.
>
> At the end of the review, there is a question: How is the exploration policy trained? We found it striking that this point was raised as the last comment in the review, in “Things to improve that did not impact the score” section. This is the fundamental research question addressed in our paper. In fact, MELEE is an algorithm that is designed to train the exploration policy for contextual bandits.
>
> Below we address the reviewer’s main questions/concerns:
>
> 1) The reviewer is concerned that our experiments show that recently published exploration methods are worse than weak baselines. (We report the best results for UCB, Cover, and Cover-Nu; see sec 5.3.) In fact, the strong performance of Eps-Greedy is expected, and has been previously reported in Bietti et al. (2018); Bastani et al. (2017). This is also discussed in the paper.
>
> 2) The reviewer does not understand the experimental setup. The experiments follow standard contextual bandit evaluation settings, and the test time behavior we follow is clearly stated and described in full details in section 3.1 and further in the evaluation methodology section in 5.2. Even further, we made the code publicly available (see footnote 1).
>
> Response to main arguments:
>
> 1) There appears to be a fundamental misunderstanding around the relationship between AggreVaTe and MELEE. It’s not true (as the review states) that “AggreVate is an instantiate [sic] of meta-learning for contextual bandits”; in fact AggreVaTe is a generic imitation learning algorithm, that we use to solve the meta-learning to explore problem. (This this stated in sec 3.1)
>
> 2) The difference between our approach and prior work: no prior work has proposed to meta-learn the exploration policy for a contextual bandit task using imitation learning (see sec 1 and the related work section 6 for a detailed comparison).
>
> 3) The variance in the results is expected in the contextual bandit setting, but accounted for: We report only statistically significant wins, and run on 300 datasets to ensure the validity of our results. As stated before, the experimental setting is fairly standard in the contextual bandit setting, and is described in sec 3.1 and 5.2.
>
> Further clarifications:
>
> 1) Many of the greek characters from the review have been deleted, so we made a best attempt to understand what the confusion is about. “F” is a functional class and is defined in sec 2: “For example, F may be a set of single layer neural networks mapping user features x \in X to predicted rewards for actions a \in [K].” The policy optimizer POLOPT is also defined in sec 2: “The oracle policy optimizer, POLOPT, takes as input a history of user interactions and outputs an f \in F with low expected regret”.
>
> 2) Thank you for pointing out the incorrect implication that the direct method is unbiased: we will rewrite this sentence to be more clear and accurate. Regardless, MELEE is agnostic to the type of optimizer used.
>
> We welcome suggestions from the reviewer for improving the clarity of the paper. To keep the paper length manageable, we found it necessary to assume that a reader has some prior familiarity with some standard ideas from imitation learning and contextual bandits. If there are aspects of the paper that could be made more clear for such readers, please let us know.
>
> Given all the clarifications above (especially the last comment in the review about training the exploration policy), it’s clear our approach was fundamentally misunderstood. We would be grateful if the reviewer would re-consider the evaluation after the clarifications highlighted above.
>
> References (also in the paper):
>
> 1) Alberto Bietti, Alekh Agarwal, and John Langford. A Contextual Bandit Bake-off. preprint, May 2018.
> 2) Bastani H, Bayati M, Khosravi K. Mostly exploration-free algorithms for contextual bandits. arXiv preprint arXiv:1704.09011. 2017 Apr 28.

---

### Author Response · Authors · 2019-11-15
**Paper Structure, Exposure, and List of Changes**

The authors appreciate the reviewers' suggestions for improving the overall exposure of the paper. In order to make it easier for reviewers’ to track the changes we kept the structure largely consistent with the original submission, but we’ll take all of these comments into account in the final version.

We've uploaded a new revision for the paper to clarify and address many of the reviewers' concerns. The list of changes in this version include:

1) Fixed reference to line 15 (was previously 16) in the theory section.
2) Fixed the incorrect implication about the bias of the direct method.
3) Clarified when exploitation happens in the paragraph describing the algorithm.
4) Mention that we use softmax to compute probabilities in the meta-features section.
5) Mention that we drop data points in the learning to rank experiments that doesn't belong to the two extremes (0, 4)

---

### Decision · Program_Chairs · 2019-12-19

**Decision:**

Reject

**Comment:**

This paper introduces MELEE, a meta-learning procedure for contextual bandits. In particular, MELEE learns how to explore by training on datasets with full-information about what every reward each action would obtain (e.g., using classification datasets). The idea is strongly related to imitation learning, and a regret bound is demonstrated for the procedure that comes from that literature. Experiments are performed.

Perhaps due to the generality in which the algorithm was presented, reviewers found some parts of the work unintuitive and difficult to follow. The work may greatly benefit from having an explicit running example for F and pi and how it evolves during training. Some reviewers were not impressed by the experimental results relative to epsilon-greedy. Yes, epsilon-greedy is a strong baseline, but MELEE introduces significant technical debt and data infrastructure so it seems fair to expect a sizable bump over epsilon-greedy or else why is it worth it?

Perhaps with revisions and experiments within a domain that justify its complexity, this paper may be suitable at another venue. But it is not deemed acceptable at this time, Reject.